# Asymptotic Behavior of Solutions to a Nonlinear Swelling Soil System with Time Delay and Variable Exponents

Mohammad M. Kafini [1,2,*], Mohammed M. Al-Gharabli [2,3] and Adel M. Al-Mahdi [2,3]

1    Department of Mathematics, King Fahd University of Petroleum and Minerals, Dhahran 31261, Saudi Arabia
2    The Interdisciplinary Research Center in Construction and Building Materials, King Fahd University of Petroleum and Minerals, Dhahran 31261, Saudi Arabia; mahfouz@kfupm.edu.sa (M.M.A.-G.); almahdi@kfupm.edu.sa (A.M.A.-M.)
3    The Preparatory Year Program, King Fahd University of Petroleum and Minerals, Dhahran 31261, Saudi Arabia
*    Correspondence: mkafini@kfupm.edu.sa

**Abstract:** In this research work, we investigate the asymptotic behavior of a nonlinear swelling (also called expansive) soil system with a time delay and nonlinear damping of variable exponents. We should note here that swelling soils contain clay minerals that absorb water, which may lead to increases in pressure. In architectural and civil engineering, swelling soils are considered sources of problems and harm. The presence of the delay is used to create more realistic models since many processes depend on past history, and the delays are frequently added by sensors, actuators, and field networks that travel through feedback loops. The appearance of variable exponents in the delay and damping terms in this system allows for a more flexible and accurate modeling of this physical phenomenon. This can lead to more realistic and precise descriptions of the behavior of fluids in different media. In fact, with the advancements of science and technology, many physical and engineering models require more sophisticated mathematical tools to study and understand. The Lebesgue and Sobolev spaces with variable exponents proved to be efficient tools for studying such problems. By constructing a suitable Lyapunov functional, we establish exponential and polynomial decay results. We noticed that the energy decay of the system depends on the value of the variable exponent. These results improve on some existing results in the literature.

**Keywords:** swelling porous problem; multiplier method; exponential and polynomial decay; time delay; variable exponents

**MSC:** 35B40; 93D20; 93D23



## 1. Introduction

The concept of time delay appears naturally when modeling some physical processes involving the displacement of a material or the transmission of energy or information. Therefore, taking this and/or such phenomena into account in the differential equations describing the evolution of such physical processes leads to more accurate models. However, it is well known that time delays may be the source of instability for some initially stable systems. In order to overcome this problem of instability, some additional damping terms should be added to the systems. In this work, we consider the following nonlinear delay swelling system:

$$
\begin{cases}
\rho_z z_{tt} - a_1 z_{xx} - a_2 u_{xx} + \xi_1 |z_t|^{m(x)-2}(x,t) z_t(x,t) + \xi_2 z_t(x,t-\tau)|z_t|^{m(\cdot)-2}(x,t-\tau) = 0, \\[2mm]
\rho_u u_{tt} - a_3 u_{xx} - a_2 z_{xx} = 0, \\[2mm]
u(x,0) = u_0(x), u_t(x,0) = u_1(x), \quad z(x,0) = z_0(x), z_t(x,0) = z_1(x) \\[2mm]
z(0,t) = z(1,t) = u(0,t) = u(1,t) = 0 \\[2mm]
z_t(x,-t) = d(x,t),
\end{cases}
\tag{1}
$$

in $(0,1) \times (0,\infty)$, where the constituents $z$ and $u$ represent the displacement of the fluid and the elastic solid material. The positive constant coefficients $\rho_u$ and $\rho_z$ are the densities of each constituent, the coefficients $a_1, a_3 > 0$ and $a_2 \neq 0$ are the coupling constants of the materials of the system, and $d$ is a history function. In addition, $\xi_1$ is a positive constant, $\xi_2$ is a real number, and $\tau$ is the time delay. By using the multiplier method, we prove that the energy of the system (1) sometimes decays exponentially and other times decays polynomially based on the value of the variable exponents. The appearance of variable exponents in the delay and damping terms in our system is important for several reasons. For example, the use of variable exponents allows for more flexible and accurate modeling of physical phenomena. They can capture the nonlinearity and heterogeneity of the damping terms, which can vary in space and time. This can lead to more realistic and precise descriptions of the behavior of fluids in different media.

Swelling soils are an environmental problem characterized by a swell in soil volume when subjected to humidity or water. The clay minerals in swelling soils attract and absorb water. When water is introduced into swelling soils, the water molecules are pulled into the gaps between the soil plates. See Figure 1 for the swelling process. As more water is absorbed, the plates are forced further apart, leading to an increase in soil pore pressure. Consequently, swelling soils lead to problems in architecture and civil engineering. They have been found all over the world. In 1997, Nelson and Miller [1] reported that the American Society of Civil Engineers estimates that one in four homes has some damage caused by swelling soils. Typically, such soils cause property owners more significant financial losses than earthquakes, floods, hurricanes, and tornadoes combined. Consequently, it is crucial to study the practical ways and means to eliminate or minimize the damages caused by swelling soils.

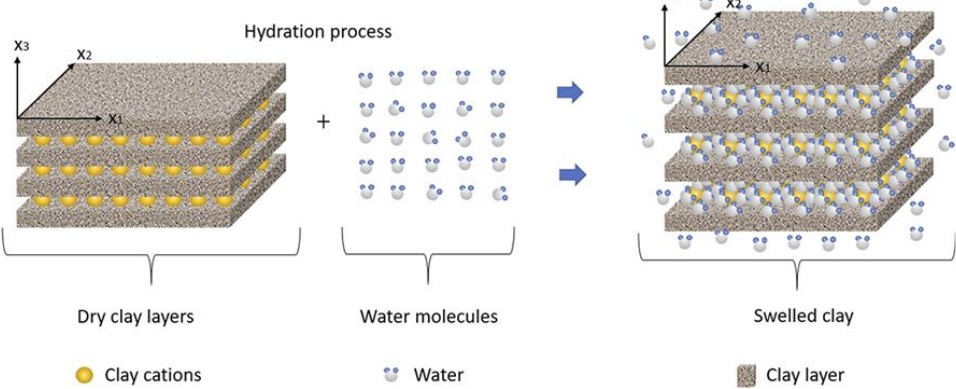

**Figure 1.** Swelling process.

The basic field equations for the linear theory of swelling porous elastic soils are mathematically given by:

$$\begin{cases} \rho_z z_{tt} = P_{1x} - I_1 + E_1 \\ \rho_u u_{tt} = P_{2x} + I_2 + E_2, \end{cases} \tag{2}$$

where the constituents $z$ and $u$ represent the displacement of the fluid and the elastic solid material, respectively. The positive constant coefficients $\rho_z$ and $\rho_u$ are the densities of the constituents $z$ and $u$, respectively. The functions $(I_1, E_1)$ represent the internal body forces and external forces acting on the displacement, respectively. This definition is the same for $(I_2, E_2)$ but acts on the elastic solid equation. The equations of the partial tensions $(P_1, P_2)$ are given by:

$$\begin{bmatrix} P_1 \\ P_2 \end{bmatrix} = \underbrace{\begin{bmatrix} a_1 & a_2 \\ a_2 & a_3 \end{bmatrix}}_{A} \begin{bmatrix} z_x \\ u_x \end{bmatrix}, \tag{3}$$

where $a_1, a_3$ are positive constants and $a_2$ is a non-zero real number, which is the coupling of the physical properties in the materials of the system. The coefficient matrix $A$ is positive definite; that is, $a_1 a_3 \geq a_2^2$. It is important to mention that the system (2) was first proposed in 1991 by Ieşan [2] and simplified in 2003 by Quintanilla [3]. The stability of the system (2) has been investigated in the literature for different damping terms. For example, Quintanilla [3] considered the following $1 - D$ swelling elastic system:

$$\begin{cases} \rho_z z_{tt} = a_1 z_{xx} + a_2 u_{xx} - \xi(z_t - u_t) + a_3 z_{xxt} & \text{in } (0, L) \times \mathbb{R}_+ \\ \rho_u u_{tt} = a_2 z_{xx} + a_3 u_{xx} + \xi(z_t - u_t), & \text{in } (0, L) \times \mathbb{R}_+, \end{cases} \tag{4}$$

where $\xi > 0$ is the constant feedback gain and $a_1 a_3 > a_2^2$. Using the energy method, the author proved that the system can be exponentially stabilized by employing three internal dampings: $u_t, z_t$, and $z_{xxt}$. Wang and Guo [4] tried to answer the same question from control theory by considering the following one-dimensional swelling elastic system:

$$\begin{cases} \rho_z z_{tt} = a_1 z_{xx} + a_2 u_{xx} - \rho_z \tau(x) z_t, \\ \rho_u u_{tt} = a_2 z_{xx} + a_3 u_{xx}, \end{cases} \tag{5}$$

where $\gamma(x)$ is an internal viscous damping function with a positive mean. Using a spectral method approach with some specific conditions on the coefficients, the authors show that the whole system can be exponentially stabilized by only one damping. After that, Ramos et al. [5], Apalara [6], and Al-Mahdi et al. [7] also considered (2) with frictional and viscoelastic damping terms, and they established explicit and general decay results under some conditions on the damping functions. For more results in porous–elasticity systems, porous–thermo-elasticity systems, porous–viscoelasticity systems, and other systems, we refer the reader to [3,4,8–19].

We notice that all the above-mentioned works are systems of partial differential equations that are independent of previous states or rates. However, systems of differential equations that are dependent on previous states are called systems of delay differential equations (DDEs). Delay differential equations are introduced to create more realistic models since many processes depend on past history. Delays rarely have an impact on how frequently the control systems operate, and delays are frequently added by sensors, actuators, and field networks that travel through feedback loops. The stability of high-speed communication networks or networked control systems are two fascinating fields of information technology and communication, where delays play a significant role. Models with delays include vehicle tracking, neural networks, population dynamics, and epidemic models, as well as sampled-data control and networked control systems, congestion control in communication networks, drilling system models, long lines with tunnel diodes, and models of lasers [20–22]. Controlling the behavior of partial differential equations' solutions with delay effects has gained scientific attention in recent years. In general, delay effects

can be found in a wide range of applications and real-world issues, including thermal, economic, chemical, biological, and physical issues. Time delays frequently have an impact on instability [23]. In recent years, there has been an increasing interest in treating equations with the variable exponent of nonlinearity. This great interest is motivated by the application to the mathematical modeling of non-Newtonian fluids. One of these fluids is the electro-rheological fluid, which has the ability to drastically change when applied to some external electromagnetic field. The variable exponent of nonlinearity is a given function of density, temperature, saturation, electric field, etc. For more information about the mathematical model of electro-rheological fluids, we refer the reader to [24,25]. The list of references concerning existence, blow-up, and stability of viscoelastic problems with variable exponents is very long, so we recall a few of them here [26–32].

To prove the stability of the system (1), we start introducing a new function (similar to [33]):

$$v(x, \rho, t) = z_t(x, t - \tau\rho), \; x \in (0,1), \; \rho \in (0,1), \; t > 0.$$

Hence, we see that $v$ satisfies

$$\tau v_t(x, \rho, t) + v_\rho(x, \rho, t) = 0, \; x \in (0,1), \; \rho \in (0,1), \; t > 0.$$

Therefore, the system (1) takes the form

$$
\begin{cases}
\rho_z z_{tt} - a_1 z_{xx} - a_2 u_{xx} + \xi_1 |z_t|^{m(x)-2}(x,t) z_t(x,t) + \xi_2 v(x,1,t)|v|^{m(\cdot)-2}(x,1,t) = 0, \\[2mm]
\rho_u u_{tt} - a_3 u_{xx} - a_2 z_{xx} = 0, \\[2mm]
\tau v_t(x, \rho, t) + v_\rho(x, \rho, t) = 0, \\[2mm]
u(x,0) = u_0(x), u_t(x,0) = u_1(x), \quad z(x,0) = z_0(x), z_t(x,0) = z_1(x), \\[2mm]
z(0,t) = z(1,t) = u(0,t) = u(1,t) = 0, \\[2mm]
v(x, \rho, 0) = d(x, -\tau\rho).
\end{cases}
\tag{6}
$$

Therefore, we consider system (6) and we prove that the system is stable under Assumptions (**A1**)–(**A3**) (below). The paper is organized as follows: In Section 2, we state some assumptions and the transformation. In Section 3, we state the main decay results, and we establish some technical lemmas in Section 4. The proofs of the stability theorems are presented in Section 5.

## 2. Preliminary and Assumptions

In this section, we consider the following **assumptions**:
(**A1**): $m : [0,1] \to [1, \infty)$ is a continuous function such that

$$m_1 := \operatorname{essinf}_{x \in [0,1]} m(x), \quad m_2 := \operatorname{esssup}_{x \in [0,1]} m(x)$$

and $1 < m_1 \leq m(x) \leq m_2 < \infty$.
(**A2**): The coefficients $a_i, \; i = 1, \ldots, 3$ satisfy $a_1 a_3 - a_2^2 > 0$.
(**A3**): The constant $\xi_1$ and $\xi_2$ satisfy the following

$$|\xi_2| < \xi_1, \tag{7}$$

and for a continuous function $\xi$,

$$\tau |\xi_2| (m(x) - 1) < \xi(x) < \tau(\xi_1 m(x) - |\xi_2|), \, x \in [0,1]. \tag{8}$$

For completeness, we state, without proof the global existence and regularity result, which can be established by a standard Galerkin argument as in [29,34].

**Theorem 1.** *Assume that conditions (A1)–(A3) hold, then problem (6) admits a unique weak solution*

$$z, u \in L^\infty\big([0,T); H_0^1(0,1)\big),$$

$$v \in L^\infty\big([0,1]; L^{m(\cdot)}((0,1) \times [0,T))\big),$$

$$z_t \in L^\infty\big([0,T); L^2(0,1)\big) \cap L^{m(\cdot)}((0,1) \times [0,T)),$$

$$u_t \in L^\infty\big([0,T); L^2(0,1)\big).$$

**Lemma 1.** *Assume that (A1)–(A3) hold. The energy of the problem (6) is defined as*

$$
\begin{aligned}
E(t) &= \frac{1}{2} \int_0^1 \Big[ \rho_z z_t^2 + \rho_u u_t^2 + a_3 u_x^2 + a_1 z_x^2 + 2a_2 z_x u_x \Big] dx \\
&\quad + \int_0^1 \int_0^1 \frac{\xi(x)|v(x,\rho,t)|^{m(x)}}{m(x)} dx d\rho,
\end{aligned}
\tag{9}
$$

*and satisfies*

$$
E'(t) \le -C_0 \left[ \int_0^1 |z_t(t)|^{m(x)} dx + \int_0^1 |v(x,1,t)|^{m(x)} dx \right] < 0.
\tag{10}
$$

**Proof.** By multiplying the first two equations of (6) by $z_t$ and $u_t$, respectively, and integrating over $(0,1)$ and multiplying the third equation of (6) by $\frac{1}{\tau}\xi(x)|v(x,1,t)|^{m(x)-2}v(x,1,t)$ and integrating over $(0,1) \times (0,1)$ using integration by parts and some manipulations, we obtain

$$
\begin{aligned}
&\frac{1}{2} \frac{d}{dt} \int_0^1 \Big[ \rho_z z_t^2 + \rho_u u_t^2 + a_3 u_x^2 + a_1 z_x^2 + 2a_2 z_x u_x \Big] + \frac{d}{dt} \int_0^1 \int_0^1 \frac{\xi(x)|v(x,\rho,t)|^{m(x)}}{m(x)} dx d\rho \\
&= -\xi_1 \int_0^1 |z_t|^{m(x)} dx - \frac{1}{\tau} \int_0^1 \int_0^1 \xi(x)|v(x,\rho,t)|^{m(x)-2} v v_\rho(x,\rho,t) d\rho dx \\
&\quad - \xi_2 \int_0^1 z_t v(x,1,t)|v(x,1,t)|^{m(x)-2} dx.
\end{aligned}
\tag{11}
$$

From (11), we see that

$$
\begin{aligned}
E'(t) &= -\xi_1 \int_0^1 |z_t|^{m(x)} dx - \frac{1}{\tau} \int_0^1 \int_0^1 \xi(x)|v(x,\rho,t)|^{m(x)-2} v v_\rho(x,\rho,t) d\rho dx \\
&\quad - \xi_2 \int_0^1 z_t v(x,1,t)|v(x,1,t)|^{m(x)-2} dx.
\end{aligned}
\tag{12}
$$

Using Young's inequality, $q = \frac{m(x)}{m(x)-1}$ and $q' = m(x)$ for the last term to obtain

$$
-\xi_2 \int_0^1 z_t v(x,1,t)|v(x,1,t)|^{m(x)-2} dx \le |\xi_2| \left( \int_0^1 \frac{1}{m(x)} |z_t|^{m(x)} dx + \int_0^1 \frac{m(x)-1}{m(x)} |v(x,1,t)|^{m(x)} dx \right).
$$

Then, (11) becomes

$$
\begin{aligned}
E'(t) \;\leq\; & -\int\limits_{0}^{1}\left[\underbrace{\xi_1 - \left(\frac{\xi(x)}{\tau m(x)} + \frac{|\xi_2|}{m(x)}\right)}_{f_1(x)>0}\right]|z_t|^{m(x)}dx \\
& -\int\limits_{0}^{1}\left[\underbrace{\frac{\xi(x)}{\tau m(x)} - \frac{|\xi_2(m(x)-1)|}{m(x)}}_{f_2(x)>0}\right]|v(x,1,t)|^{m(x)}dx.
\end{aligned}
\tag{13}
$$

Since $m(x)$, then $\xi(x)$ is bounded. Hence, we infer that $f_1(x)$ and $f_2(x)$ are also bounded. So, if we define $C_0(x) = \min\{f_1(x), f_2(x)\} > 0$ for any $x \in [0,1]$, and take $C_0(x) = \inf_{[0,1]}\{C_0(x)\}$, so $C_0(x) \geq C_0 > 0$. Moreover, by using assumptions (**A1**)–(**A2**), the proof of (10) is completed. $\square$

## 3. The Main Results

In this section, we state our decay results.

**Theorem 2.** *Assume that (**A1**)–(**A3**) hold, and $1 < m_1 < 2$. Then, the energy functional (9) satisfies for a positive constants C, depends on $m_1$,*

$$
E(t) < C\left(\tfrac{1}{t+1}\right)^{\left(\frac{m_1-1}{2-m_1}\right)}, \; \forall \, t > 0. \;\cdot
\tag{14}
$$

**Theorem 3.** *Assume that (**A1**)–(**A3**) hold. Then, the energy functional (9) satisfies for two positive constants $\lambda_1$, $\lambda_2$ and for any $t \geq 0$,*

$$
E(t) \leq \lambda_1 e^{-\lambda_2 t}, \quad \text{if } m_1 = m_2 = 2,
\tag{15}
$$

*and*

$$
E(t) \leq \lambda_1 \left(\frac{1}{t+1}\right)^{\left(\frac{2}{m_2-2}\right)}, \quad \text{if } m_1, m_2 > 2.
\tag{16}
$$

## 4. Technical Lemmas

In this part, we state and prove some needed lemmas.

**Lemma 2.** *For any $\eta > 0$ and $m_1 \geq 2$, we have the following*

$$
\int_0^1 z|z_t|^{m(\cdot)-2}z_t dx \leq c_1\eta \int_0^1 z_x^2 dx + \int_0^1 c_\eta(x)|z_t|^{m(x)}dx.
\tag{17}
$$

*and if $1 < m_1 < 2$, we have*

$$
\int_0^1 z|z_t|^{m(\cdot)-2}z_t dx \leq 2c\eta||z_x||_2^2 + c_\eta\left[\int_0^1 |z_t|^{m(x)}dx + \left(\int_0^1 |z_t|^{m(x)}\right)^{m_1-1}dx\right].
\tag{18}
$$

**Lemma 3.** *For any $\gamma > 0$ and $m_1 \geq 2$, we have the following*

$$
\int_0^1 u(x,t)|v(x,1,t)|^{m(\cdot)-2}v(x,1,t)dx \leq c_2\gamma \int_0^1 u_x^2 dx + \int_0^1 c_\gamma(x)|v(x,1,t)|^{m(x)}dx.
\tag{19}
$$

*and if* $1 < m_1 < 2$, *we have*

$$\int_0^1 u|v(x,1,t)|^{m(\cdot)-2}v(x,1,t)dx \quad \leq c_2\gamma \int_0^1 u_x^2 dx + \int_0^1 c_\gamma(x)|v|^{m(x)}dx$$

$$+ \left( \int_0^1 |v|^{m(x)} \right)^{m_1-1} dx. \tag{20}$$

**Proof.** We will prove Lemma 2 and the proof of Lemma 3 will be the same. We start by applying Young's inequality with with $p(x) = \frac{m(x)}{m(x)-1}$ and $p'(x) = m(x)$. So, for a.e $x \in [0,1]$ and any $\delta_1 > 0$, we have

$$|z_t|^{m(x)-2}z_t z \leq \delta_1|z|^{m(x)} + c_{\delta_1}(x)|z_t|^{m(x)},$$

where

$$c_{\delta_1}(x) = \delta_1^{1-m(x)}(m(x))^{-m(x)}(m(x)-1)^{m(x)-1}.$$

Hence,

$$\int_0^1 z|z_t|^{m(x)-2}z_t dx \leq \delta_1 \int_0^1 |z|^{m(x)}dx + \int_0^1 c_{\delta_1}(x)|z_t|^{m(x)}dx. \tag{21}$$

Next, using (9) and (10), Poincaré's inequality and the embedding property, we get

$$
\begin{aligned}
\int_0^1 |z|^{m(x)}dx &= \int_{\Omega_+} |z|^{m(x)}dx + \int_{\Omega_-} |z|^{m(x)}dx \\
&\leq \int_{\Omega_+} |z|^{m_2}dx + \int_{\Omega_-} |z|^{m_1}dx \\
&\leq \int_0^1 |z|^{m_2}dx + \int_0^1 |z|^{m_1}dx \\
&\leq c_e^{m_1}||z_x||_2^{m_1} + c_e^{m_2}||z_x||_2^{m_2} \\
&\leq \left( c_e^{m_1}||z_x||_2^{m_1-2} + c_e^{m_2}||z_x||_2^{m_2-2} \right)||z_x||_2^2 \\
&\leq \left( c_e^{m_1}\left(\frac{2}{a_1}E(0)\right)^{m_1-2} + c_e^{m_2}\left(\frac{2}{a_1}E(0)\right)^{m_2-2} \right)||z_x||_2^2 \\
&\leq c_1||z_x||_2^2,
\end{aligned}
\tag{22}
$$

where $c_e$ is the embedding constant,

$$\Omega_+ = \{x \in [0,1] : |z(x,t)| \geq 1\}, \quad \Omega_- = \{x \in [0,1] : |z(x,t)| < 1\}$$

and

$$c_1 = \left( c_e^{m_1}\left(\frac{2}{a_1}E(0)\right)^{m_1-2} + c_e^{m_2}\left(\frac{2}{a_1}E(0)\right)^{m_2-2} \right). \tag{23}$$

Then, (21) and (22) yield

$$\int_0^1 z|z_t|^{m(x)-2}z_t dx \leq \delta_1 c_1||z_x||_2^2 + \int_0^1 c_{\delta_1}(x)|z_t|^{m(x)}dx. \tag{24}$$

Hence, the proof of (17) is completed. To prove (18), we set

$$\Omega_1 = \{x \in [0,1] : m(x) < 2\} \text{ and } \Omega_2 = \{x \in [0,1] : m(x) \geq 2\}.$$

Then, we have

$$\int_0^1 z|z_t|^{m(x)-2}z_t dx = -\int_{\Omega_1} z|z_t|^{m(x)-2}z_t dx - \int_{\Omega_2} z|z_t|^{m(x)-2}z_t dx. \tag{25}$$

We notice that on $\Omega_1$, we have

$$2m(x) - 2 < m(x), \text{ and } 2m(x) - 2 \geq 2m_1 - 2. \tag{26}$$

Therefore, by using Young's and Poincaré's inequalities and (26) leads to

$$
\begin{aligned}
-\int_{\Omega_1} z|z_t|^{m(x)-2}u_t dx &\leq \eta \int_{\Omega_1} |z|^2 dx + \frac{1}{4\eta}\int_{\Omega_1}|z_t|^{2m(x)-2}dx \\
&\leq \eta\|z_x\|_2^2 + c_\eta\left[\int_{\Omega_1^+}|z_t|^{2m(x)-2}dx + \int_{\Omega_1^-}|z_t|^{2m(x)-2}dx\right] \\
&\leq \eta\|z_x\|_2^2 + c_\eta\left[\int_{\Omega_1^+}|z_t|^{m(x)}dx + \int_{\Omega_1^-}|z_t|^{2m_1-2}dx\right] \\
&\leq \eta\|z_x\|_2^2 + c_\eta\left[\int_0^1|z_t|^{m(x)}dx + \left(\int_{\Omega_1^-}|z_t|^2 dx\right)^{m_1-1}\right] \\
&\leq \eta\|z_x\|_2^2 + c_\eta\left[\int_0^1|z_t|^{m(x)}dx + \left(\int_{\Omega_1^-}|z_t|^{m(x)}dx\right)^{m_1-1}\right] \\
&\leq \eta\|z_x\|_2^2 + c_\eta\left[\int_0^1|z_t|^{m(x)}dx + \left(\int_0^1|z_t|^{m(x)}dx\right)^{m_1-1}\right],
\end{aligned} \tag{27}
$$

where

$$\Omega_1^+ = \{x \in \Omega_1 : |z_t(x,t)| \geq 1\} \text{ and } \Omega_1^- = \{x \in \Omega_1 : |z_t(x,t)| < 1\}. \tag{28}$$

Next, we have, by the case $m(x) \geq 2$,

$$\int_{\Omega_2} z|z_t|^{m(x)}z_t dx \leq \eta\|z_x\|_2^2 + \int_0^1 c_\eta(x)|z_t|^{m(x)}dx. \tag{29}$$

Combining (25)–(29), so estimate (18) is established. □

**Lemma 4.** *Assume that (A1)–(A3) hold. The functional*

$$\chi_1(t) = -\rho_u\varepsilon\int_0^1 u_t u\, dx \tag{30}$$

*satisfies, for $0 < \varepsilon < 1$, $\varepsilon_1 > 0$, and $\bar{c} > 0$ depends on $a_1, a_2, a_3, \alpha$,*

$$\chi_1'(t) \leq -\rho_u\varepsilon\int_0^1 u_t^2 dx + \bar{c}\varepsilon\varepsilon_1\int_0^1 u_x^2 dx + \frac{\bar{c}\varepsilon}{\varepsilon_1}\int_0^1 z_x^2 dx. \tag{31}$$

**Proof.** Differentiating $\chi_1$ and using (6) gives

$$\chi_1'(t) = -\rho_u\varepsilon\int_0^1 u_t^2 dx + a_3\varepsilon\int_0^1 u_x^2 dx + a_2\varepsilon\int_0^1 u_x z_x dx. \tag{32}$$

Using Young's inequality, we have for $\varepsilon_1 > 0$,

$$a_2\int_0^1 u_x z_x dx \leq \varepsilon_1\int_0^1 u_x^2 dx + \frac{a_2^2}{4\varepsilon_1}\int_0^1 z_x^2 dx. \tag{33}$$

Combining the above estimate, then the proof of (31) is completed. □

**Lemma 5.** *Assume that (A1)–(A3) hold. The functional*

$$\chi_2(t) = a_2\rho_z \int_0^1 uz_t\, dx - a_2\rho_u \int_0^1 zu_t\, dx \tag{34}$$

*satisfies, for any $\varepsilon_2 > 0$ and $\bar{c} > 0$,*

$$
\begin{aligned}
\chi_2'(t) \;\leq\; & \varepsilon_2 \int_0^1 u_t^2 dx + \frac{\bar{c}}{\varepsilon_2} \int_0^1 z_t^2 dx + \bar{c} \int_0^1 z_x^2 dx - \frac{a_2^2}{2} \int_0^1 u_x^2 dx \\
& + c \int_0^1 |z_t|^{m(x)} dx + c \int_0^1 |v(x,1,t)|^{m(x)} dx.
\end{aligned} \tag{35}
$$

**Proof.** By exploiting (6), we have

$$
\begin{aligned}
\chi_2'(t) \;=\; & a_2\rho_z \int_0^1 u_t z_t\, dx - a_2\rho_u \int_0^1 u_t z_t\, dx - a_2 a_1 \int_0^1 u_x z_x\, dx - a_2^2 \int_0^1 u_x^2 dx \\
& - a_2\xi_1 \int_0^1 u|z_t|^{m(\cdot)-2} z_t dx - a_2\xi_2 \int_0^1 u|v(x,1,t)|^{m(\cdot)-2} v(x,1,t) dx \\
& + a_2 a_3 \int_0^1 z_x u_x dx + a_2^2 \int_0^1 z_x^2 dx.
\end{aligned}
$$

Using Young's inequality, we have for $\varepsilon_2 > 0$,

$$a_2(\rho_z - \rho_u) \int_0^1 u_t z_t dx \leq \varepsilon_2 \int_0^1 u_t^2 dx + \left( \frac{a_2^2 \rho_z^2}{4\varepsilon_2} + \frac{a_2^2 \rho_u^2}{4\varepsilon_2} \right) \int_0^1 z_t^2 dx. \tag{36}$$

Similarly,

$$a_2(a_3 - a_1) \int_0^1 u_x z_x dx \leq a_2^2 \eta_2 \int_0^1 u_x^2 dx + \frac{(a_1^2 + a_3^2)}{4\eta_2} \int_0^1 z_x^2\, dx. \tag{37}$$

Using (17) and (19), we have

$$I_1 := -a_2\xi_1 \int_0^1 u|z_t|^{m(\cdot)-2} z_t dx \leq a_2\xi_1 c_1 \eta \int_0^1 u_x^2 dx + \int_0^1 c_\eta(x)|z_t|^{m(x)} dx, \tag{38}$$

and

$$
\begin{aligned}
I_2 : \;\; & = -a_2\xi_2 \int_0^1 u(x,t)|v(x,1,t)|^{m(\cdot)-2} v(x,1,t) dx \\
& \leq c_2 a_2 |\xi_2| \gamma \int_0^1 u_x^2 dx + \int_0^1 c_\gamma(x)|v(x,1,t)|^{m(x)} dx.
\end{aligned} \tag{39}
$$

Combining (38) and (47), choosing $\gamma = \eta$, $c_0 = \min\{c_1\xi_1, c_2\xi_1\}$, we obtain

$$I_1 + I_2 \leq c_0 \eta \int_0^1 u_x^2 dx + \int_0^1 c_\eta(x)|z_t|^{m(x)} dx + \int_0^1 c_\gamma(x)|v(x,1,t)|^{m(x)} dx. \tag{40}$$

Choosing $\eta = \frac{a_2^2}{2c_0}$ and combining all the above estimations, then the proof of (35) is completed. $\square$

**Lemma 6.** *Assume that (A1)–(A3) hold. The functional*

$$\chi_3(t) \quad = \rho_z \int_0^1 zz_t\, dx - \frac{a_2}{a_3} \rho_u \int_0^1 u_t z\, dx \tag{41}$$

*satisfies, for any $\varepsilon_3 > 0$ and $\bar{c} > 0$,*

$$
\begin{aligned}
\chi_3'(t) \leq & -\frac{\alpha_0}{2} \int_0^1 z_x^2 dx + \frac{\bar{c}}{\varepsilon_3} \int_0^1 z_t^2 dx + \varepsilon_3 \int_0^1 u_t^2 dx \\
& + c \int_0^1 |z_t|^{m(x)} dx + c \int_0^1 |v(x,1,t)|^{m(x)} dx.
\end{aligned}
\tag{42}
$$

*where $\alpha_0 = a_1 - \frac{a_2^2}{a_3} > 0$.*

**Proof.** In view of (6) and integration by parts, we obtain

$$
\begin{aligned}
\chi_3'(t) = & \rho_z \int_0^1 z_t^2 \, dx - \left[ a_1 - \frac{a_2^2}{a_3} \right] \int_0^1 z_x^2 \, dx - \frac{a_2}{a_3} \rho_u \int_0^1 u_t z_t dx \\
& \underbrace{-\xi_1 \int_0^1 z |z_t|^{m(x)-2} z_t dx}_{I_3} \underbrace{-\xi_2 \int_0^1 z|v(x,1,t)|^{m(x)-2} v(x,1,t) dx}_{I_4}.
\end{aligned}
\tag{43}
$$

Using Young's inequality, we have for $\varepsilon_3 > 0$,

$$
-\frac{a_2}{a_3} \rho_u \int_0^1 u_t z_t dx \leq \varepsilon_3 \int_0^1 u_t^2 dx + \frac{a_2^2}{a_3^2 \varepsilon_3} \rho_u^2 \int_0^1 z_t^2 dx.
\tag{44}
$$

Using (17) and (19), we have

$$
I_3 \leq \xi_1 c_1 \eta \int_0^1 z_x^2 dx + \int_0^1 c_\eta(x) |z_t|^{m(x)} dx,
\tag{45}
$$

and

$$
I_4 \leq |\xi_2| c_1 \gamma \int_0^1 z_x^2 dx + \int_0^1 c_\gamma(x) |v(x,1,t)|^{m(x)} dx.
\tag{46}
$$

Combining (45) and (46), choosing $\gamma = \eta$, we obtain

$$
I_3 + I_4 \leq c_1 \xi_1 \eta \int_0^1 z_x^2 dx + \int_0^1 c_\eta(x) |z_t|^{m(x)} dx + \int_0^1 c_\gamma(x) |v(x,1,t)|^{m(x)} dx.
\tag{47}
$$

Inserting the last estimates in (43), choosing $\eta = \frac{\alpha_0}{c_1 \xi_1}$, the proof of (42) is completed. $\square$

**Lemma 7.** *Assume that (A1)–(A3) hold. The functional*

$$
\chi_4(t) = -\rho_z \int_0^1 z z_t dx
\tag{48}
$$

*satisfies, for some $\varepsilon > 0$,*

$$
\begin{aligned}
\chi_4'(t) \leq & -\rho_z \int_0^1 z_t^2 dx + \varepsilon_4 \int_0^1 u_x^2 dx + \frac{c}{\varepsilon_4} \int_0^1 z_x^2 dx \\
& + c \int_0^1 |z_t|^{m(x)} dx + c \int_0^1 |v(x,1,t)|^{m(x)} dx.
\end{aligned}
\tag{49}
$$

**Proof.** Direct computations, using (6), give

$$
\begin{aligned}
\chi_4'(t) = & -\rho_z \int_0^1 z_t^2 \, dx + a_1 \int_0^1 z_x^2 dx + a_2 \int_0^1 u_x z_x dx \\
& + \xi_1 \int_0^1 z |z_t|^{m(x)-2} z_t dx + \xi_2 \int_0^1 z |v(x,1,t)|^{m(x)-2} v(x,1,t) dx.
\end{aligned}
\tag{50}
$$

Repeating the same above estimations, the estimates (52) and (52) are established. □

**Lemma 8.** *Assume that (A1)–(A3) hold. The functional*

$$\chi_5(t) = \tau \int_0^1 \int_0^1 e^{-\rho\tau}\xi(x)|v(x,\rho,t)|^{m(x)}dxd\rho, \tag{51}$$

*satisfies,*

$$\chi_5'(t) \leq \int_0^1 \xi(x)|z_t(t)|^{m(x)}dx - \tau e^{-\tau}\int_0^1\int_0^1 \xi(x)|v(x,\rho,t)|^{m(x)}dxd\rho. \tag{52}$$

**Proof.** The proof of the above lemma can be found in many papers in the literature; see, for instance, Lemma 4.2 in [35]. □

**Lemma 9.** *Assume that (A1)–(A3) hold. Then*

$$\int_0^1 z_t^2 dx \leq -cE'(t), \quad \text{if } m_1 = m_2 = 2, \tag{53}$$

*and*

$$\int_0^1 z_t^2 dx \leq -cE'(t) + c\left(-E'(t)\right)^{\frac{2}{m_2}}, \quad \text{if } m_1, m_2 > 2. \tag{54}$$

**Proof.** By recalling (10), it is easy to establish (53). To prove (54), we set the following partitions

$$\Omega_1 = \{x \in [0,1] : |z_t| \geq 1\} \quad \text{and} \quad \Omega_2 = \{x \in [0,1] : |z_t| < 1\}. \tag{55}$$

With use of Hölder's and Young's inequalities and (9), we obtain

$$\int_{\Omega_1} z_t^2 dx \leq \int_0^1 |z_t|^{m(x)}dx = -cE'(t), \tag{56}$$

$$\begin{aligned}
\int_{\Omega_2} z_t^2 \mathrm{d}x &\leq c\left(\int_{\Omega_2}|z_t|^{m_2}\mathrm{d}x\right)^{\frac{2}{m_2}} \\
&\leq c\left(\int_{\Omega_2}|z_t|^{m(x)}\mathrm{d}x\right)^{\frac{2}{m_2}} \leq c\left(\int_0^1|z_t|^{m(x)}\mathrm{d}x\right)^{\frac{2}{m_2}} = c\left(-E'(t)\right)^{\frac{2}{m_2}}. \tag{57}
\end{aligned}$$

Combining (56) and (57), estimate (54) is established. □

## 5. Proofs of Theorems 2 and 3

In what follows, we prove Theorem 2.

**Proof.** Let

$$\mathcal{L}(t) = \mu E(t) + \mu_1\chi_1(t) + \mu_2\chi_2(t) + \mu_3\chi_3(t) + \mu_4\chi_4(t) + \chi_5(t) \tag{58}$$

where $\mu, \mu_1, \mu_2, \mu_3, \mu_4$ are positive constants to be properly chosen. By taking the derivative of the functional $\mathcal{L}$ and using all the above estimates (31)–(52), we obtain

$$
\begin{aligned}
\mathcal{L}'(t) \leq &-\left(\frac{a_2^2}{2}\mu_2 - \bar{c}\varepsilon\varepsilon_1\mu_1 - \varepsilon_4\mu_4\right)\int_0^1 u_x^2 dx \\
&-\left(\frac{\alpha_0}{2}\mu_3 - \frac{\bar{c}\varepsilon\mu_1}{\varepsilon_1} - \bar{c}\mu_2 - \frac{\bar{c}\mu_4}{\varepsilon_4}\right)\int_0^1 z_x^2 \, dx \\
&-\left(\mu_4 - \frac{\bar{c}\mu_2}{\varepsilon_2} - \frac{\bar{c}\mu_3}{\varepsilon_3}\right)\rho_z\int_0^1 z_t^2 dx \\
&-\left(\varepsilon\mu_1 - \varepsilon_2\mu_2 - \varepsilon_3\mu_3\right)\int_0^1 u_t^2 dx \\
&-\left[C_0\mu - c\mu_2 - c\mu_3 - c\mu_4 - c\right]\int_0^1 |z_t|^{m(\cdot)} dx \\
&-\left[C_0\mu - c\mu_2 - c\mu_3 - c\mu_4\right]\int_0^1 |v(x,1,t)|^{m(\cdot)} dx \\
&-\tau e^{-\tau}\int_0^1\int_0^1 \xi(x)|v(x,\rho,t)|^{m(x)} dx d\rho \\
&+c\left(\int_0^1 |z_t|^{m(x)} dx\right)^{m_1-1} + c\left(\int_0^1 |v(x,1,t)|^{m(x)} dx\right)^{m_1-1} dx. \quad (59)
\end{aligned}
$$

Choosing $\varepsilon_i = \mu_i$, $i = 1,2,3,4$, then the above estimate becomes

$$
\begin{aligned}
\mathcal{L}'(t) \leq &-\left(\frac{a_2^2}{2}\mu_2 - \bar{c}\varepsilon\mu_1^2 - \mu_4^2\right)\int_0^1 u_x^2 dx \\
&-\left(\frac{\alpha_0}{2}\mu_3 - \bar{c}\varepsilon\mu_1^2 - \bar{c}\mu_2 - c\right)\int_0^1 z_x^2 \, dx \\
&-(\mu_4 - 2\bar{c})\rho_z\int_0^1 z_t^2 dx \\
&-\left(\varepsilon\rho_u\mu_1 - \mu_2^2 - \mu_3^2\right)\int_0^1 u_t^2 dx \\
&-\left[C_0\mu - c\mu_2 - c\mu_3 - c\mu_4 - c\right]\int_0^1 |z_t|^{m(\cdot)} dx \\
&-\left[C_0\mu - c\mu_2 - c\mu_3 - c\mu_4\right]\int_0^1 |v(x,1,t)|^{m(\cdot)} dx \\
&-\tau e^{-\tau}\int_0^1\int_0^1 \xi(x)|v(x,\rho,t)|^{m(x)} dx d\rho \\
&+c\left(\int_0^1 |z_t|^{m(x)} dx\right)^{m_1-1} + c\left(\int_0^1 |v(x,1,t)|^{m(x)} dx\right)^{m_1-1} dx. \quad (60)
\end{aligned}
$$

First, we select $\mu_4$ such that

$$
\mu_4 - 2\bar{c} > 1.
$$

Then, we choose a $\mu_2$ large enough such that

$$
\Lambda_1 := \frac{a_2^2}{2}\mu_2 - \mu_4^2 > 0.
$$

Then, we choose a $\mu_3$ large enough such that

$$
\Lambda_2 := \frac{\alpha_0}{2}\mu_3 - \bar{c}\mu_2 - c > 0.
$$

Then, we choose a $\mu_1$ large enough such that

$$
\varepsilon\rho_u\mu_1 - \mu_2^2 - \mu_3^2 > 1.
$$

Now, we set

$$\varepsilon \leq \min\{\frac{\Lambda_1}{\bar{c}\mu_1^2}, \frac{\Lambda_2}{\bar{c}\mu_1^2}\},$$

so that

$$\Lambda_1 - \bar{c}\mu_1^2 > 0$$

and

$$\Lambda_2 - \bar{c}\mu_1^2 > 0.$$

After fixing $\mu_i$, where $i = 1, 2, 3, 4$, we select a $\mu$ large enough (if needed) such that

$$C_0\mu - c\mu_2 - c\mu_3 - c\mu_4 - c > 0,$$

and $\mathcal{L} \sim E$. That is, we can find two positive constants $\alpha_1$ and $\alpha_2$ such that

$$\alpha_1 E(t) \leq \mathcal{L}(t) \leq \alpha_2 E(t). \tag{61}$$

On the other hand, Young's inequality and (9) allow us to conclude that

$$E(t) \leq \bar{c} \int_0^1 \left( u_t^2 + u_x^2 + z_t^2 + z_x^2 \right) dx + c \int_0^1 \int_0^1 \xi(x)|v(x,\rho,t)|^{m(x)} dx d\rho. \tag{62}$$

Hence, estimate (60) becomes for any $t \geq 0$ and for some positive constant $\alpha_3$,

$$\begin{aligned}
\mathcal{L}'(t) \quad &\leq \quad -\alpha_3 \int_0^1 \left( u_t^2 + u_x^2 + z_t^2 + z_x^2 + \int_0^1 \xi(x)|v(x,\rho,t)|^{m(x)} dx d\rho \right) dx \\
&+ \quad c \left( \int_0^1 |z_t|^{m(x)} dx \right)^{m_1-1} + c \left( \int_0^1 |v(x,1,t)|^{m(x)} dx \right)^{m_1-1} dx.
\end{aligned} \tag{63}$$

Then, from (63) and (62), we arrive at

$$\mathcal{L}'(t) \leq -\alpha_4 E(t) + c \left( \int_0^1 |z_t|^{m(x)} dx \right)^{m_1-1} + c \left( \int_0^1 |v(x,1,t)|^{m(x)} dx \right)^{m_1-1} dx, \quad \forall t \geq 0, \tag{64}$$

and thanks to (61), we obtain, for any $t \geq 0$,

$$\mathcal{L}'(t) \leq -\alpha_5 \mathcal{L}(t) + c \left( \int_0^1 |z_t|^{m(x)} dx \right)^{m_1-1} + c \left( \int_0^1 |v(x,1,t)|^{m(x)} dx \right)^{m_1-1} dx.$$

Using (10), multiplying the above equation by $E^\alpha(t)$, $\alpha = \frac{2-m_1}{m_1-1}$, using the fact $E \sim \mathcal{L}$, and using Young's inequality, we obtain:

$$\begin{aligned}
E^\alpha(t)\mathcal{L}'(t) &\leq -\alpha_5 E^{\alpha+1}(t) + cE^\alpha(t)\left( -E'(t) \right)^{m_1-1} \\
&\leq -\alpha_5(1-\varepsilon)E^{\alpha+1}(t) + \frac{c}{\varepsilon}\left( -E'(t) \right).
\end{aligned} \tag{65}$$

Taking a $\varepsilon$ small enough and using the non-increasing property of $E$, (65) becomes:

$$\mathcal{L}_1(t) \leq -\alpha_6 E^{\alpha+1}(t), \qquad \forall t \geq 0, \tag{66}$$

where $\mathcal{L}_1(t) = E^\alpha(t)\mathcal{L}(t) + cE(t)$.

Integration over $(0,t)$, using $E \sim \mathcal{L}_1$, gives

$$E(t) \leq c_{m_1} \left( \frac{1}{t+1} \right)^{\frac{1}{\alpha}}, \ \forall\, t > 0, \tag{67}$$

where $\alpha = \frac{2-m_1}{m_1-1} > 0$. Then, the proof of (14) is completed. $\square$

Now, to prove (3), we reformulate the integral $\int_0^1 z_t^2 dx$ in (60) and recall that the integrals $\left( \int_0^1 |z_t|^{m(x)} dx \right)^{m_1-1}$ and $\left( \int_0^1 |v(x,1,t)|^{m(x)} dx \right)^{m_1-1}$ are not involved in this case, so we have

**Proof.**

$$
\begin{aligned}
\mathcal{L}'(t) \leq\ & -\left( \frac{a_2^2}{2}\mu_2 - \bar{c}\varepsilon\mu_1^2 - \mu_4^2 \right) \int_0^1 u_x^2 dx \\
& - \left( \frac{\alpha_0}{2}\mu_3 - \bar{c}\varepsilon\mu_1^2 - \bar{c}\mu_2 - c \right) \int_0^1 z_x^2\, dx \\
& - \mu_4 \rho_z \int_0^1 z_t^2 dx + 2\bar{c}\rho_z \int_0^1 z_t^2 dx \\
& - \left( \varepsilon\rho_u\mu_1 - \mu_2^2 - \mu_3^2 \right) \int_0^1 u_t^2 dx \\
& - [C_0\mu - c\mu_2 - c\mu_3 - c\mu_4 - c] \int_0^1 |z_t|^{m(\cdot)} dx \\
& - [C_0\mu - c\mu_2 - c\mu_3 - c\mu_4] \int_0^1 |v(x,1,t)|^{m(\cdot)} dx \\
& - \tau e^{-\tau} \int_0^1 \int_0^1 \xi(x)|v(x,\rho,t)|^{m(x)} dx d\rho
\end{aligned}
\tag{68}
$$

First, we choose $\mu_4 > 0$ any real number; then, we do the same selections (as in the above proof). By recalling Poincaré's inequality and (10), estimate (68) becomes, for a positive constant $\beta_3$,

$$
\mathcal{L}'(t) \leq -\beta_3 \int_0^1 \left( u_t^2 + u_x^2 + z_t^2 + z_x^2 + \int_0^1 \xi(x)|v(x,\rho,t)|^{m(x)} dx d\rho \right) dx + \int_0^1 z_t^2 dx, \quad \forall t \geq 0.
\tag{69}
$$

In the other hand, Young's inequality and (9) allow us to conclude that

$$
E(t) \leq \bar{c} \int_0^1 \left( u_t^2 + u_x^2 + z_t^2 + z_x^2 + \int_0^1 \xi(x)|v(x,\rho,t)|^{m(x)} dx d\rho \right) dx.
\tag{70}
$$

Then, from (69) and (70), we arrive at

$$
\mathcal{L}'(t) \leq -\beta_4 E(t) + \int_0^1 z_t^2 dx, \quad \forall t \geq 0,
\tag{71}
$$

and thanks to (61), we obtain, for any $t \geq 0$,

$$
\mathcal{L}'(t) \leq -\beta_5 \mathcal{L}(t) + \int_0^1 z_t^2 dx.
$$

Now, we will discuss two cases:
**Case 1**: if $m_2 = 2$, then by using Lemma 9, we have

$$
\mathcal{L}'(t) \leq -\beta_5 \mathcal{L}(t) + (-E'(t)).
$$

This gives

$$
\mathcal{L}_1'(t) \leq -\beta_5 \mathcal{L}(t).
$$

where $\mathcal{L}_1 = (\mathcal{L} + E) \sim E$. Integrating the last estimate over the interval $(0,t)$ and using the equivalence properties $\mathcal{L}_1, \mathcal{L} \sim E$, the proof of (15) is completed.

**Case 2**: if $m_2 > 2$, then by using Lemma 9, we have

$$\mathcal{L}'(t) \leq -\beta_5 \mathcal{L}(t) + (-E'(t))^{\frac{2}{m_2}}.$$

Multiplying the last equation by $E^q$, where $q = \frac{m_2 - 2}{2}$, then we obtain

$$E^q \mathcal{L}'(t) \leq -\beta_5 E^q \mathcal{L}(t) + E^q(-E'(t))^{\frac{2}{m_2}}.$$

With use of Young's inequality with $\gamma = \frac{q+1}{q}$ and $\gamma* = \frac{m_2}{m_2 - 2}$, we obtain for $\varepsilon > 0$

$$E^q \mathcal{L}'(t) \leq -(\beta_5 - \varepsilon)E^{q+1}\mathcal{L}(t) + C_\varepsilon(-E'(t)).$$

Taking a $\varepsilon$ small enough and using the non-increasing property of $E$, the above estimate becomes:

$$\mathcal{L}_2(t) \leq -\beta_6 E^{q+1}(t), \qquad \forall t \geq 0, \tag{72}$$

where $\mathcal{L}_2 = E^q \mathcal{L} + cE \sim E$.

Integration over $(0, t)$, using $E \sim \mathcal{L}_2$, gives

$$E(t) < c_{m_2}\left(\frac{1}{t+1}\right)^{1/q}, \forall\, t > 0, \tag{73}$$

where $q = \frac{m_2 - 2}{2} > 0$. Then, the proof of (17) is completed. $\square$

## 6. Conclusions

We considered a nonlinear swelling soil system with a time delay and nonlinear damping with a variable exponent type. We used the multiplier method to investigate the long-term behavior of this system, and we obtained exponential and polynomial decay results under some conditions on the variable exponent. Our results are established under the more general assumption of the damping; we considered a nonlinear damping of variable exponent type, so these results generalize and extend to several results in the literature. We believe that our results will contribute to better understanding and mastering the issue of stabilization by nonlinear damping with a variable exponent and will probably add to the implementation of these types of dissipations in damping technology. Our results will certainly allow a wider class of frictional damping functions to be used for stabilization and be a basis for further work.

**Author Contributions:** Conceptualization, M.M.K., M.M.A.-G. and A.M.A.-M.; methodology, M.M.K., M.M.A.-G. and A.M.A.-M.; validation, M.M.K., M.M.A.-G. and A.M.A.-M.; formal analysis M.M.K., M.M.A.-G. and A.M.A.-M.; investigation, M.M.K., M.M.A.-G. and A.M.A.-M.; writing—original draft preparation, A.M.A.-M.; writing—review and editing, M.M.A.-G. and M.M.K.; visualization, A.M.A.-M.; supervision, M.M.A.-G.; project administration, A.M.A.-M.; funding acquisition, M.M.A.-G. All authors have read and agreed to the published version of the manuscript.

**Funding:** This research was funded by King Fahd University of Petroleum and Minerals (KFUPM-IRC-CBM) grant number INCB2315.

**Data Availability Statement:** No data were used to support this study.

**Acknowledgments:** The authors would like to thank King Fahd University of Petroleum and Minerals (KFUPM) for its support. The authors also thank the referees for their very careful reading and valuable comments. This was funded by KFUPM, Grant No. INCB2315.

**Conflicts of Interest:** The authors declare that there is no conflict of interest.

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
