# Peer review of "Asymptotic Behavior of Solutions to a Nonlinear Swelling Soil System with Time Delay and Variable Exponents"

_mca, doi:10.3390/mca28050094_

Round 1

Reviewer 1 Report

see the report.

Reviewer 2 Report

In this paper, the authors considered the asymptotic behavior of a nonlinear swelling soil system with time delay and variable exponents on the damping and on the delay term. By constructing a suitable Lyapunov functional, they established exponential and polynomial decay results. 

The problems studied are meaningful and the results seem to be correct. But I think there are some inappropriate points in this paper. Here are some suggestions:

(1)   In the article, the format of many formulas is not neat, such as (2.5), (4.4), (4.11) and so on.

(2)   The authors obtain at least a polynomial decay result when 1 < m1 < 2. I'm interested in whether this result is optimal, right? That is, whether the authors can show the lack of exponential stability?

In addition, there are many grammatical errors throughout the manuscript, and the authors should double-check and correct them. There is just a small part of the mistakes in the paper. Of course, there are more than these in the article.

(1)   In line 12 on page 1, “instability some” should be “instability, some”;

(2)   In line 19 on page 2, “soils are” should be “soil is”;

(3)   In line 37 on page 2, “Similar definition be the same for (I2, E2) but acting on the elastic solid equation.” maybe have mistakes in grammar;

(4)   In line 13 on page 3, “try” should be “tried”;

(5)   In line 2 on page 4, “system” should be “systems”;

(6)   In line 5 on page 4, “were” should be “are”;

(7)   In line 1 on page 5, “System” should be “system”;

(8)   In line 18 on page 5, “Problem” should be “problem”;

(9)   In line 15 on page 6, “Since m(x), and hence ξ(x), is bounded, we infer that f1(x) and f2(x) are also bounded.” is not properly expressed;

(10) In line 1 on page 9, “and (4.10) lead” should be “, (4.10) leads”;

(11) In line 17 on page 9, “Direct computations, using (1.6), give” is not properly expressed;

(12) In line 21 on page 9, “os” should be “of”;

(13) In line 18 on page 10, “Choosing” should be “and choosing”;

(14) In line 23 on page 10, “os” should be “of”.

There are many grammatical errors throughout the manuscript, and the authors should double-check and correct them.

Author Response

Dear Editor: Thank you and the referees for the valuable and very helpful report

on our manuscript. We went over the referee's  comments carefully

and made all the suggestions. We updated the abstract and conclusion accodingly. We believe that the revised version is

much better and is worthy of consideration for publication.

Reviewer 3 Report

The paper introduces several novelties to the area of swelling soil systems, both in terms of the mathematical techniques employed and the specific features of the model. These innovations have the potential to enhance our understanding of swelling soils and could pave the way for more accurate predictions and mitigation strategies.

The use of DDEs in modelling swelling soil systems appears to be a significant novelty. Many processes in nature and engineering depend on past states or rates, and by introducing DDEs, the authors aim to create a more realistic model that accounts for these delays. This can be particularly relevant in scenarios where the past state of the soil has an impact on its current behaviour.

The paper makes several assumptions regarding the continuous function “m” and the coefficients “ai”. While assumptions are necessary for mathematical modelling, it's crucial to ensure they don't overly simplify real-world conditions. The authors could provide more justification for these assumptions or discuss their implications in real-world scenarios.

I recommend that the authors provide a more detailed description of the paper's novelties in the abstract. Currently, the abstract appears somewhat ambiguous. While it's unusual to see equations within the abstract, it might be more effective to discuss the equation's significance there and present the actual equation later in the paper.

The conclusions could benefit from clearer articulation, as they currently come across as both vague and overly concise. Typically, conclusions transition from specific points to broader implications. I encourage the authors to enhance the presentation of the paper's general contributions.

Finally, I understand that the paper shows some novelties to the field but is mainly focused on the mathematical aspects without addressing the practical implications or potential applications of their findings. 

The English writing of the article is not a problem; the English is appropriate

Round 2

Reviewer 3 Report

The authors have made the suggested revisions and improved the article's version. I would still recommend to the authors that they enhance the abstract and conclusion, as I believe that providing more information about the applicability of the developed model would attract more readers' attention to the article. However, I believe the paper is suitable for publication with the aforementioned revisions. I want to thank the authors for incorporating the suggested revisions.

Author Response

(The authors gave the same response as above.)
